# Fermented Foods and Their Role in Respiratory Health: A Mini-Review

**Periyanaina Kesika** [1,2], **Subramanian Thangaleela** [1], **Bhagavathi Sundaram Sivamaruthi** [2,*], **Muruganantham Bharathi** [1] and **Chaiyavat Chaiyasut** [1,*]

1   Innovation Center for Holistic Health, Nutraceuticals and Cosmeceuticals, Faculty of Pharmacy, Chiang Mai University, Chiang Mai 50200, Thailand; p.kesika@gmail.com (P.K.); leelasubramanian@gmail.com (S.T.); bharathi.m03@gmail.com (M.B.)
2   Office of Research Administration, Chiang Mai University, Chiang Mai 50200, Thailand
*   Correspondence: sivamaruthi.b@cmu.ac.th (B.S.S.); chaiyavat@gmail.com (C.C.); Tel.: +66-53-944-340 (C.C.)

**Abstract:** Fermented foods (FFs) hold global attention because of their huge advantages. Their health benefits, palatability, preserved, tasteful, and aromatic properties impart potential importance in the comprehensive evaluation of FFs. The bioactive components, such as minerals, vitamins, fatty acids, amino acids, and other phytochemicals synthesized during fermentation, provide consumers with several health benefits. Fermentation of food is an ancient process that has met with many remarkable changes owing to the development of scientific technologies over the years. Initially, fermentation relied on back-slapping. Nowadays, starter cultures strains are specifically chosen for the type of fermentation process. Modern biotechnological methods are being implemented in the fermentation process to achieve the desired product in high quality. Respiratory and gastrointestinal tract infections are the most severe health issues affecting human beings of all age groups, especially children and older adults, during this COVID-19 pandemic period. Studies suggest that the consumption of probiotic *Lactobacillus* strains containing fermented foods protects the subjects from common infectious diseases (CIDs, which is classified as upper respiratory tract infections, lower respiratory tract infections and gastrointestinal infections) by improving the host's immune system. Further studies are obligatory to develop probiotic-based functional FFs that are effective against CIDs. Presently, we are urged to find alternative, safe, and cost-effective prevention measures against CIDs. The current manuscript briefs the production of FFs, functional properties of FFs, and their beneficial effects against respiratory tract infections. It summarizes the outcomes of clinical trials using human subjects on the effects of supplementation of FFs.

**Keywords:** fermented foods; probiotics; *Lactobacillus*; respiratory infection; immune system; traditional fermentation

## 1. Introduction

Fermented foods state that "foods that have been transformed due to fermentation process via microbes such as bacteria, yeast, fungi and their enzymes". The word fermentation is derived from the Latin word "*fervere*" meaning "to boil" [1] or "fermentare" meaning "to leaven" [2]. By simple means, fermented foods (FFs) are microbiologically processed raw materials (vegetables, meats, etc.). The International Scientific Association for Probiotics and Prebiotics defined FFs as "foods made through desired microbial growth and enzymatic conversions of food components" [3]. FFs are mainly available in the form of pickles, fermented juices, dairy, and meat products. The use of FFs varies depending on the availability of raw materials, region, and ethnicity [4]. FFs and beverages are enzymatically altered by the microorganisms in such a way that they taste and smell appealing to humans for consumption [5]. In addition, this improves the palatability of foods and insists on preservation [6]. The traditional preparation of FFs involves the natural fermentation process. The microorganisms responsible for the fermentation process

define the products' quality and type. Lactic acid bacteria (LAB), *Bacillus* spp., yeast, and molds are the commonly found critical microbes in fermented food products [7]. The FFs have been used to improve health status. Even though FFs have a long history, recently, due to increased attention from biologists, technologists, and consumers, many studies have been engaged in demonstrating that FFs can enhance gastrointestinal and systemic health [8–11]. The beneficial effects of the continuous consumption of FFs on inflammatory bowel disease [12], immunity boost [13], oxidative stress [14], cardiovascular diseases [15] cognitive enhancement [16], anxiety [17], and metabolic disorders [18] have been previously described. Nutritional intervention with fermented foods enhances immune function and alleviates upper respiratory tract infections (URTIs) [19], as well as improving intestinal and extra-intestinal health [20]. Zhang et al. [21] reported that daily consumption of Qingrun yogurt (fermented milk) for 12 weeks reduces the common cold in adults living in Northern China [21]. Ingestion of LAB-fermented food modulates the gut microbiome through their health promoting properties and production of metabolites such as biopeptides, vitamins, organic acids, bacteriocins, and anti-microbial compounds during fermentation [22].

Respiratory tract infections, such as URTIs and lower respiratory tract infections (LRTI), are a universal health threat among developed and developing countries [23]. LRTI, such as pneumonia, can be fatal in young and elderly individuals [24]. Viruses are responsible for causing more than 90% of URTIs [25]. The only way to prevent RTIs is to improve the immune function [19]. URTIs are infection in the mucosal surfaces of upper airways, such as the nose, sinuses, pharynx, or larynx, resulting in non-allergic rhinitis, acute sinusitis, acute pharyngitis/laryngitis, acute epiglottis, and acute otitis media [26–28]. Some of the viruses responsible for common respiratory infections are the influenza virus, respiratory syncytial virus, parainfluenza virus, rhinovirus, enterovirus, adenovirus, coronavirus, and others [28–30]. The influenza virus causes serious health threats, which may lead to morbidity [31]. The recent pandemic SARS-COV-2 virus and its associated respiratory infections create an alarming call about strengthening immune responses against a wide range of organisms that cause RTIs [32]. Acute respiratory infections in children is a serious health threat across the world, which also results in increased morbidity and mortality [33]. In this COVID-19 pandemic situation, respiratory and gastrointestinal tract infections are the most severe health issues affecting human beings of all age groups, especially children. Several treatments are available to treat respiratory infection [34]. However, a well-balanced diet, preferably comprised of probiotics or fermented functional foods, is also crucial for maintaining body homeostasis and fighting against respiratory and alimentary tract viruses. The probiotics in FFs enhance antiviral activity by producing bioactive compounds. These bioactive compounds improve immune function and lessen viral infections [35]. Naturally, the human body encompasses mechanical defense systems against the respiratory tract viruses, such as mucus, ciliated cells, and macrophages, which ward off the virus particles entering the system [36,37]. The LAB in FFs exhibit antiviral activity against the respiratory tract virus [38]. Certain FFs have proven to have anti-influenza functions. For example, kimchi, which contains *Lactobacillus casei* DK128, possesses antiviral activity against the influenza H3N2 virus [39]. The intervention of *Lactobacillus rhamnosus* GG in fermented milk product for 3 months demonstrated the prevention of respiratory discomforts in children attending daycare centers [40]. Probiotic delivery through fermentation is a combined approach; thus, probiotics in fermented foods produces added benefits and is advantageous over non-fermented products [41,42].

The current review summarizes the results of scientific reports on the beneficial effects of the consumption of FFs on respiratory infections and briefs the traditional and modern aspects of the production of FFs, functional properties of FFs, and their beneficial effects against respiratory tract infection. It summarizes the outcomes of clinical trials using human subjects on the effects of the supplementation of FFs. Peer-reviewed papers published in scientific journals were searched by using the keywords "Fermented foods", "Respiratory infection", "fermented food production", "starter culture", "probiotics in food", and "health benefits of fermented foods" in PubMed, Scopus, Web of Science, and Google Scholar. The

relevant papers were systemically screened and used to prepare the manuscript. Articles not published in the English language were not included in the study.

## 2. Traditional and Modern Aspects of the Production of FFs

Fermented food products, such as wine, beer, dairy products, and baked foods, are emerging globally and date back to 13,000 BC [43]. Fermentation can be categorized into indigenous fermentation (FFs produced traditionally by spontaneous fermentation) used for small scale production and technological fermentations (FFs produced using modern technological facilities) used for industrial scale production [2,43]. Extended shelf life, food safety, and improved organoleptic properties, such as aroma, taste, and texture, are important factors for fermentation practices [44]. The FFs play a prominent role in the socio-economic status of developing countries. The fermentation involves various biochemical processes by microorganisms and enzymes, resulting in significant food modification. The evolution of fermentation has been initiated ever since the beginning of human civilization. The traditional fermentation method is uncontrolled, as it depends on microorganisms from the environment [45]. The traditional fermentation process is of great economic value and reasonable food preservation method. Traditionally, the fermentation of foods occurs through two methods. Natural fermentation of foods and beverages are known as 'wild ferments' [46], where fermentation occurs by naturally available autochthonous or indigenous microbiota present in the food raw materials, which was subjected to changes depending on the surrounding environmental conditions, resulting in the characteristic FFs. The FFs are specific according to their geographic locations [43,47]. The second method is called 'culture-dependent ferments' [46–51]. Every new batch of fermentation was started with back-slopping, where a small amount of successful previous fermented batch product was used as an inoculum or natural starter culture [48].

The nutritional health benefits, increased food safety, and organoleptic features lead to the growing demand for FFs. Meanwhile, this leads to the increased awareness of food safety measures, standardization measures, industrial control of food production, starter cultures, and fermentation process control etiquettes at the industrial level [43]. The demand for FFs and increasing customers leads to an industrial standardization of fermentation. The first and prudent step is the selection of starter cultures, which is very important in determining the total fermentation process, their safety, organoleptic properties, and necessary changes in the food substrates [49]. From identifying various fermentation microbes and starter cultures, the progression of the development of fermentation techniques becomes pertinent. The current development in molecular biology techniques, next-generation sequencing (NGS), multi-omics and bioinformatics tools, and highly advanced statistical tools leads to further advancements, such as studying the genome sequence of industrially important microbes, their diversity, functions, and metabolic pathways. These outcomes result in great progress in the fermentation industry [50].

The term starter culture usually represents the prepared inoculum containing viable microorganisms that belongs to one or two or more than the two selected microbial species or strains. Thus, the desirable changes in the final product is obtained by using these selected commercial starter cultures while processing the raw materials to initiate the fermentation process. The purpose of starter cultures in fermentation is to fasten the process and to produce characteristic fermented foods. These starter cultures are specifically identified and isolated with the help of advanced microbiological techniques and are currently used in the industrial fermentation process to produce beverages such as beer, alcohol, and vinegar, and foods such as bread and other meat, as well as dairy products [52]. Such inoculation with selected microbes minimizes the risk of foodborne diseases [53,54]. The starter cultures had also been selected from the screened isolates, which yielded end products with high organoleptic features in previous fermentation processes [55]. Molecular approaches, such as high-throughput screening (HTS) of target genes, genetic engineering of specific and well-adapted starter cultures, are synthesized and incorporated to create better-improved fermentation [52].

The starter cultures are improved through another novel technology, CRISPR/Cas9 (Clustered Regularly Interspaced Short Palindromic Repeats/CRISPR-associated protein 9), where the genome of the specific target microbe is precisely edited [56]. Thus, the microbiome manipulation using CRISPR-based technologies improves specific genes in starter cultures [57]. Advanced genetic engineering can rule out undesirable microbial populations and enhance the desirable microbiota, leading to optimized FFs [58]. Recent scientific advances could aid in developing the whole food microbial ecosystem, their interactions, and pathways owing to desirable fermentation processes [59]. Despite the desired quality of starter cultures produced through recombinant DNA technology, those strains, and their genetically modified food ingredients are restricted from usage due to stern food regulations and a lack of acceptance from consumers [60]. Various other technologies include dominant selection, random mutagenesis, natural competence, and conjugation [61].

In the traditional method, the microbial composition of the fermented products depends on the microbial culture used. Outdated plate cultivation method is unable to provide accurate microbial profile information; henceforth, recent NGS technology encompassing collective studies of metagenomics, meta-transcriptomics, meta-proteomics, and metabolomics enables the identification of accurate microbiome from different microbial communities, which pave the way to the detailed study of microbe–microbe interactions in fermentation ecosystems [62]. FFs contain different types of microbial ecosystems such as bacteria, fungi, and yeast; altogether, they are responsible for the quality and safety of FFs [63]. Integrated multi-omics technologies, next-generation nucleic acid, and protein studies enhance the microbes used for the fermentation process, and provide improved fermented products with prolonged shelf life and desirable characteristics. The stability and safety of fermentation and FFs have been improved. The genome editing and molecular approaches rapidly improved the microbial functions, resulting in better quality products with increased safety and an extended shelf life. The next-generation nucleic acid and protein-based approaches provide adequate details about the microbial community for better fermentation and desired products with more shelf life [64].

## 3. Functional Properties of Fermented Food and Beneficial Effects against Respiratory Tract Infection

By definition, FFs can be either plant or animal-based and are produced through back-slopping or by spontaneous fermentation via enzymatic actions of microorganisms so that raw materials are converted into FFs. FFs are rich in nutrients with better taste and provide health benefits [65,66]. Every so often, FFs and beverages are supplemented with probiotics to improve nutritional and health properties [67]. FFs are familiar because of their health benefits and safety standards. Not all FFs have live organisms; in some products, they have been heat-treated, inactivating the microbes. Moreover, FFs are a rich source of bioactive microbes [14]. FFs are functional foods that generally boost the immune functions and metabolism of the consumer [51]. Probiotics are used to ferment the foods traditionally, hence, probiotics and fermented foods are closely related [68]. During fermentation, various bioactive peptides, oligosaccharides, lipids, and other components are synthesized. These components improve the functionality of the product, such as antioxidants, antihypertensive, and other bioactivities [69–71]. The health benefits of FFs include preventing metabolic diseases, cardiovascular diseases, enhancing immune function, and improving cognition [12,14,15,17,72]. In 1907, Metchnikoff explained the health benefits of fermented milk products [73]. The traditional fermented milk products are highly rich in bioactive compounds such as peptides, amino acids, vitamins, and minerals [74]. Bioactive compounds are found in unprocessed counterparts such as organic acids, fatty acids, bacteriocins, amino acids, and exopolysaccharides [75,76]. Naturally, bioactive compounds are being released into the system in two different ways. One is that it is released from raw materials because of high acidity. Another way is due to the hydrolyzing enzymes of the probiotic microflora [77]. The FFs are involved in the

improvement of intestinal permeability and barrier function [78] and demonstrate positive effects on atherosclerosis, inflammatory bowel diseases, colon cancer [79], anger, depression, and anxiety [80]. FFs aid in maintaining the gut microflora with the presence of bioactive chemicals, neuropeptides, antioxidants, and anti-inflammatory activity [81].

Probiotic bacteria bind directly to the virus and disrupt the adhesion of virus cells onto the mucosal cells [82]. Muhialdin et al. reviewed the connection between FFs, their probiotics with the anti-viral mechanisms and the immune system, and proficient antiviral activity against respiratory and alimentary tract ailments [35]. The antiviral mechanisms involve various systemic changes such as enhancing natural killer cells (NK cells), pro-inflammatory cytokines, and cytotoxic T lymphocytes (CD3$^+$, CD16$^+$, and CD56$^+$) [83]. Bioactive yogurt compounds resist upper respiratory tract infections [84–88] and remain potentially beneficial by preventing the common cold and influenza [89,90].

Gouda et al. [91] reviewed a few studies [92–95] and stated that yogurt bioactive peptides and probiotics exhibited antiviral properties against respiratory infectious viruses that share some mechanistic similarities with the SARS-CoV-2 virus [91]. Cell-free supernatants of the metabolites of yogurt fermented with *Lactobacillus plantarum* and yogurt fermented with *Bifidobacterium bifidum* demonstrated anti-enterovirus 71 activity [92]. Metabolites of both *L. casei*, and *Bifidobacterium adolescentis* demonstrated effective antiviral activity against a rotavirus infection [93]. In vitro studies by Starosila et al. [94] described that the influenza virus was inhibited completely by the P18 peptide produced by *Bacillus subtilis*. Similarly, another research study [95] in mice stated that *Lactobacillus gasseri* SBT2055 prevents the influenza A virus and respiratory syncytial viral infections [95]. As per the review provided by Gouda et al. [91], the yogurt bioactive peptides are effective against the COVID-19 virus through the inhibition of the angiotensin-converting enzyme (ACE) and might influence the COVID-19 presentation and outcome [91].

Korean traditionally fermented foods that contain grains, fruits, herbs, and mushrooms provide high levels of *Lactobacilli* that undergo lactic acid fermentation and produce different metabolites that exhibit beneficial health effects [96] such as crucial antiviral properties against the influenza virus [97]. Probiotic LAB or the fermented foods containing probiotic LAB elicit an antagonist effect on the influenza virus by stimulating immune responses by increasing interferon (IFN-α, IFN-β, and IFN-γ) and interleukin (IL-2, IL-6, and IL-12) levels [98]. *L. plantarum* (YML009) [99] and *L. plantarum* (DK119) [97] isolated from kimchi were found to have antiviral activity against the H1N1 influenza virus [97,99]. The yogurt containing *L. lactis* reduces influenza symptoms [100]. Japanese-fermented foods containing *L. plantarum* YU mitigate influenza virus replication [101]. *L. plantarum* DK119 from kimchi was proven to demonstrate complete protection against influenza-A viruses [97]. Oral administration of fermented yoghurt containing *Lactobacillus bulgaricus* OLL1073R-1 or its exopolysaccharides ameliorates the influenza viral infection by developing splenocytic NK cell activity [102]. *L. plantarum* and *Leuconosoc mesenteroides* demonstrated antiviral activity against the H1N1 and H1N9 influenza virus [103].

The supplementation of 65 mL of probiotic fermented milk containing *L. rhamnosus* GG (ATCC 53103) ($7.1 \times 10^9$ CFU per day), *Bifidobacterium* sp. B420 ($8.4 \times 10^9$ CFU per day), *Lactobacillus acidophilus* 145 ($3.2 \times 10^9$ CFU per day), and *Streptococcus thermophilus* ($27 \times 10^9$ CFU per day) to human volunteers (patients with nasal potentially pathogenic bacteria; $n = 108$; 33 females, 75 males; age = $41 \pm 8$ years old) for 28 days reduced the colonization of potent nasal pathogenic bacteria (*Staphylococcus aureus*, *Streptococcus pneumoniae*, and β-hemolytic streptococci) compared to that of the control group (patients with nasal potentially pathogenic bacteria; $n = 101$; 29 females, 72 males; age = $39 \pm 9$ years old) supplemented with standard yogurt (180 g containing conventional LAB strains such as *Lactobacillus delbrueckii* subsp. *bulgaricus* and *S. thermophilus*; total $\geq 10^7$ CFU per gram). The study proposed that the oral consumption of probiotic fermented milk may stimulate the B lymphocytes of gut-associated lymphoid tissue, which may further reach the upper respiratory system and elucidate secretory IgA production [104].

Shift-based workers (*n* = 500; 43% males; age = 31.8 ± 8.9 years old) [84] and free-living elderly (*n* = 537; 339 females, 198 males; age range = 72–80 years old) [105] were supplemented with a fermented dairy product (Actimel®; dosage 200 g per day) containing the probiotic *L. casei* DN-114001 (≥ $10^{10}$ CFU/100 g of product) and the common starter culture *L. delbrueckii* subsp. *bulgaricus* and *S. thermophilus* (total CFU of starter cultures ≥ $10^9$ CFU per 100 g) for three months, and the effects on common respiratory and gastrointestinal infectious diseases (CIDs) were monitored [84,105]. The total study period includes 2 weeks of diet control prior to probiotic supplementation, 3 months of probiotic supplementation and a 1-month follow-up period without probiotic supplementation [84,105]. The probiotic supplementation reduced the incidence of CIDs, cumulated duration of fever, cumulated number of CIDs (in the subgroup of smokers), and increased the time to the first incidence of CIDs in the probiotic group. The probiotic supplementation improved the immune cell (leukocyte, neutrophil, and natural killer) counts and its activity in shift-based workers compared to that of the control group (*n* = 500; 44% males; age = 32.5 ± 8.9 years old). The study indicated that the daily consumption of Actimel® could reduce the incidence and severity of CIDs in stressed individuals [84]. Similarly, the supplementation of Actimel® significantly reduced the average period per episode and cumulative duration of CIDs in elderly subjects compared to the placebo control group (*n* = 535; 333 females, 202 males; age range = 73–81 years old) supplemented with non-fermented dairy product. The probiotic-supplemented group significantly reduced both the cumulative durations and episodes of URTIs (upper respiratory tract infections) and rhinopharyngitis. The fecal load of *L. casei* was also found to be high in the probiotic group. The study proved that the fermented product was safe, and regular consumption of fermented products made with the probiotic *L. casei* DN-114 001 could reduce the incidence of CIDs, especially URTIs, in elderly subjects [105]. Healthy children (*n* = 314; 157 females, 157 males; age range = 3 to 6 years old) attending daycare centers/schools were supplemented with Actimel® for 3 months, and the incidence of CIDs and behavioral changes due to illness were assessed. The intervention with Actimel® for three months effectively reduced the incidence of CIDs in children with no differences in behavioral changes due to illness compared to that of the control group (*n* = 324; 152 females, 172 males; age range = 3 to 6 years old) supplemented with the non-fermented acidified dairy product [106].

Healthy children (*n* = 139; 61 females, 78 males; age range = 13 to 86 months old) attending daycare centers were supplemented with 100 mL of fermented milk product containing $10^9$ CFU of the probiotic *L. rhamnosus* strain GG (LGG) for three months. The study demonstrated that the LGG supplementation significantly reduced the risk, duration, and symptoms of URTIs compared to that of the placebo group (*n* = 142; 63 females, 79 males; age range = 13 to 83 months old) supplemented with post-pasteurized fermented milk product without LGG. No changes were observed in the incidence and duration of gastrointestinal infections, vomiting, and diarrheal episodes among the study subjects. Thus, daily intake of LGG reduces the URTIs in children attending daycare centers [40].

Fujita et al. [85] investigated the effect of daily consumption of fermented milk (dosage 80 mL per day) containing the probiotic *L. casei* strain Shirota (4 × $10^{10}$ CFU/80 mL of product) on URTIs in elderly (*n* = 76; 57 females, 19 males; age = 83.5 ± 8.9 years old) at daycare facilities. The daily intake of probiotic reduced the duration of infection in elderly compared to that of the placebo group (*n* = 78; 52 females, 26 males; age = 83.0 ± 9.3 years old) supplemented with the placebo drink (same drink without probiotic *L. casei* strain Shirota) [85].

Makino et al. [89] evaluated the effect of the intake of yoghurt (dosage: 90 g per day; for 8 weeks at the Funagata study or for 12 weeks at the Arita study) fermented with the probiotic *L. delbrueckii* subsp. *bulgaricus* OLL1073R-1 (2.0–3.5 × $10^8$ CFU per gram) and *S. thermophilus* OLS3059 (6.3–8.8 × $10^8$ CFU per gram) on the common cold in healthy elderly subjects. In the Funagata study, 57 elderlies were selected (age range = 69–80 years old) and assigned for the probiotic group (*n* = 29) and milk group (*n* = 28). In the Arita study, 101 elderlies were selected (age range = 59–84 years old) and assigned for the probiotic

group ($n$ = 43) and milk group ($n$ = 42). Both studies in the probiotic group exhibited reduction in the risk of catching the common cold compared to that of the milk group supplemented with milk (dosage 100 mL per day for 8 or 12 weeks). The probiotic intake also increased the quality of life score for the eye/nose/throat system, which was correlated with the improved natural killer cell activity in elderly compared to that of the milk group [89].

Healthy children (1–4 years old) attending daycare or preschool were supplemented with fermented cow's milk (FCM) containing *Lactobacillus paracasei* CBA L74 ($5.9 \times 10^9$ CFU/g) in powder form (the live microbes are heat-killed before spray drying the fermented milk product) (daily dosage: 7 g of powder diluted in 150 mL of cow's milk or water) for three months, and the incidence of CIDs and changes in immunity was assessed. The FCM supplementation significantly reduced the incidence of CIDs and URTIs and decreased acute gastroenteritis in children of the probiotic group ($n$ = 66) compared to that of the placebo group ($n$ = 60) supplemented with maltodextrins. Changes in the innate and acquired immune system were also recorded in FCM-treated groups. The study concluded that FCM containing the heat-killed CBA L74 could improve their immunity in children [107]. Another study by Nocerino et al. also reported that the supplementation of FCM and fermented rice product containing *L. paracasei* CBA L74 ($5.9 \times 10^9$ CFU per gram; heat-killed) prevents CIDs' incidence by prompting the innate and acquired immune system in children (age = $23 \pm 10$ months old; FCM group $n$ = 137; fermented rice product group $n$ = 118) compared to that of the placebo group ($n$ = 122) supplemented with maltodextrins [108].

A twelve-week intervention with fermented milk containing *L. casei* strain Shirota ($1.0 \times 10^{11}$ CFU per day) significantly reduced the incidence, duration, and symptoms of URTIs in middle-aged (30–49 years old) office workers of the probiotic group ($n$ = 49) compared to that of the control group ($n$ = 47) supplemented with milk. The study also revealed that the probiotic supplementation activated the NK cells. Both the control and probiotic group exhibited increased salivary cortisol secretion. The results suggest that the daily consumption of fermented milk containing *L. casei* strain Shirota may reduce the risk of URTIs in middle-aged office workers [109]. Mai et al. reported that consuming fermented milk containing *L. casei* strain Shirota ($1.0 \times 10^8$ CFU per ml; dosage = 65 mL per day) for 12 weeks reduces the incidence of diarrhea, constipation, and acute respiratory tract infection in children ($n$ = 510; 236 females, 274 males; age range = 3–5 years old) compared to that of the control group ($n$ = 493; 209 females, 284 males; age range = 3–5 years old) [110] (Table 1).

**Table 1.** The reported beneficial effect of FFs against respiratory tract infection.

| S. No. | Type of Study | Fermented Product | Active Microbes | Subjects/ Model | Key Findings | Reference |
|---|---|---|---|---|---|---|
| 1 | Randomized, double-blind, and controlled trial. | Fermented milk containing probiotic | *L. rhamnosus* GG | Healthy children | ↓ Risk of URTIs ↓ Duration of infection | [40] |
| 2 | Randomized, double-blind, and controlled trial. | fermented dairy drink containing probiotic (Actimel®) | Probiotic: *L. casei* DN-114 001; Conventional cultures: *L. delbrueckii* subsp. *bulgaricus* and *S. thermophilus* | Shift-based workers | ↓ Incidence of CIDs ↑ Neutrophil, leukocyte, and NK cell count and activity | [84] |
| 3 | Multicenter, randomized, double-blind, placebo-controlled trial. | Fermented milk containing probiotic | *L. casei* strain Shirota | Elderly subjects | ↓ Duration of URTIs | [85] |
| 4 | Randomized, double-blind and placebo-controlled trial. | Fermented yoghurt containing probiotic | Probiotic: *L. delbrueckii* subsp. *bulgaricus* OLL1073R-1; starter culture: *S. thermophilus* | Elderly subjects | ↑ NK cell activity ↓ Risk of the common cold | [89] |

**Table 1.** *Cont.*

| S. No. | Type of Study | Fermented Product | Active Microbes | Subjects/Model | Key Findings | Reference |
|---|---|---|---|---|---|---|
| 5 | Open, prospective trial | Fermented milk containing probiotics | *Lactobacillus* GG (ATCC 53103), *Bifidobacterium* sp. B420, *Lactobacillus acidophilus* 145, and *S. thermophilus* | Patients carrying potentially pathogenic bacteria in their nasal cavity | ↓ Nasal pathogenic bacteria (↓ Gram +[ve] bacteria) | [104] |
| 6 | Randomized, double-blind, and controlled trial. | Fermented dairy drink containing probiotic (Actimel®) | Probiotic: *L. casei* DN-114 001; starter cultures: *L. delbrueckii* subsp. *bulgaricus* and *S. thermophilus* | Elderly subjects | ↓ Incidence of CIDs ↓Durations and episodes of URTIs | [105] |
| 7 | Randomized, double-blind, and controlled trial. | Fermented dairy drink containing probiotic (Actimel®) | *L. casei* DN-114 001; Starter cultures: *L. delbrueckii* subsp. *bulgaricus* and *S. thermophilus* | Healthy children | ↓ Incidence of CIDs | [106] |
| 8 | Randomized, double-blind, and controlled trial. | Fermented cow's milk | *L. paracasei* CBA L74 (heat-killed) | Healthy children | ↓ Acute gastroenteritis ↓ URTIs | [107] |
| 9 | Randomized, double-blind, and controlled trial. | Fermented cow's milk and fermented rice product | *L. paracasei* CBA L74 (heat-killed) | Healthy children | ↓ Incidence of CIDs ↓ URTIs; ↑ immunity | [108] |
| 10 | Randomized, double-blind, and controlled trial. | Fermented milk containing probiotic | *L. casei* strain Shirota | Healthy office workers | ↓ Incidence and symptoms of URTIs ↑ Salivary cortisol levels in both probiotic and control group ↓ Constipation | [109] |
| 11 | Controlled filed trial | Fermented milk containing probiotic | *L. casei* strain Shirota | Children | ↓ Incidence of diarrhea ↓ Acute respiratory infection | [110] |

CIDs: Respiratory and gastrointestinal common infectious diseases; NK: Natural killer; URTIs: Upper respiratory tract infections.

## 4. Conclusions

The daily consumption of probiotic-based fermented products could improve the host's health status and aid in resisting respiratory infections. Advanced and powerful biotechnological tools are used to calibrate and standardize fermentation processes and provide more desirable fermented foods with enhanced quality, taste, aroma, long shelf life, nutritional content, and safety, and prevent contamination by spoilage organisms. Fermentation has undergone remarkable changes over the years. Genome editing of the microbiome through NGS and CRISPR/Cas9 techniques are efficiently employed in fermentation industries to improve the functionality of the microbial community in different cultures. In addition to the wide range of applications of fermentation in the food industry, this technology was also found to be prominent in treating some diseases such as immune, respiratory, gastrointestinal diseases, and common infectious diseases. A minimal number of clinical trials have been conducted using human subjects on the effects of fermented food supplementation on CIDs.

FFs used in all the clinical trials that are included in this review resulted to be probiotics supplemented with fermented dairy products that exhibited beneficial effects against respiratory tract infections in children, adults, and elderly individuals. Among the reviewed studies reporting the probiotics used in the intervention of FFs, *L. casei* [84,85,105,106,109,110] and *L. paracasei* (heat-killed) [107,108] are the most commonly used probiotics for health-promoting properties. The results supported the assertion that regular consumption of probiotic-containing fermented drinks could reduce the risk of CIDs [84,105–108,110] and URTIs [40,84,85,89,104,105,107–110], especially in children [40,106–108,110]. The possible mechanisms behind the protective effect of probiotic-based fermented products are the stimulation of the host's innate and acquired immune system. The studies suggested that

the consumption of probiotic *Lactobacillus* strains containing fermented foods protected the subjects from CIDs and upper respiratory tract infections by improving the host's immune system. However, many clinical trial studies with fermented food products with antiviral properties are further required to know the complete mechanisms of the probiotic FF products for its use, in order to develop conventional probiotic-based functional foods that are effective against respiratory tract infections.

**Author Contributions:** Conceptualization, B.S.S., P.K. and C.C.; methodology, P.K. and S.T.; validation, B.S.S., S.T., P.K. and C.C.; formal analysis, S.T. and B.S.S.; investigation, S.T. and B.S.S.; resources, C.C.; writing—original draft preparation, S.T., B.S.S., P.K. and C.C.; writing—review and editing, S.T., B.S.S., P.K., M.B. and C.C.; supervision, C.C. and B.S.S.; project administration, C.C.; funding acquisition, C.C. All authors have read and agreed to the published version of the manuscript.

**Funding:** This research received funding from the Chiang Mai University, Thailand.

**Institutional Review Board Statement:** Not applicable.

**Informed Consent Statement:** Not applicable.

**Data Availability Statement:** Not applicable.

**Acknowledgments:** We wish to thank the Faculty of Pharmacy, Chiang Mai University, Chiang Mai, Thailand. The study was partially supported by the Chiang Mai University, Chiang Mai, Thailand.

**Conflicts of Interest:** The authors declare no conflict of interest.

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
