# Peer review of "Fermented Foods and Their Role in Respiratory Health: A Mini-Review"

_fermentation, doi:10.3390/fermentation8040162_

Round 1

Reviewer 1 Report

The authors should reorganise the the whole manuscript. For example, part 6 (line 197), part 7 (line 253) and part 8 (line 287) can be merge into one part. Also, I do not understand the necessity of part 5 (line 166). More Figures and Tables should be included to help readers understand the manuscript. Strain names should be italic (line 239, line 560, line 595 et al.,). 

Author Response

Reviewer: 1

Comment: The authors should reorganise the whole manuscript. For example, part 6 (line 197), part 7 (line 253) and part 8 (line 287) can be merge into one part. Also, I do not understand the necessity of part 5 (line 166).

Author response: The whole manuscript has been reorganized in the revised manuscript. As per Reviewer suggestions, part 6, 7 and 8 has been merged into one part in the revised manuscript. Part 5 (and references cited in part 5 and figure 1) have been removed in the revised manuscript.

Comment: More Figures and Tables should be included to help readers understand the manuscript.

Author response: The whole manuscript has been reorganized and detailed for better clarification in the revised manuscript.

Since the intervention and wash-out period is not mentioned clearly (different period mentioned in the abstract and methodology) in the reference article “Marotta, F.; Naito, Y.; Jain, S.; Lorenzetti, A.; Soresi, V.; Kumari, A.; Carrera Bastos, P.; Tomella, C.; Yadav, H. Is there a potential application of a fermented nutraceutical in acute respiratory illnesses? An in-vivo placebo-controlled, cross-over clinical study in different age groups of healthy subjects. J. Biol. Regul. Homeost. 2012, 26, 285-294.”, It has been removed from the table and text of the revised manuscript.

Comment: Strain names should be italic (line 239, line 560, line 595 et al.,).

Author response: Complied in the revised manuscript. 

Reviewer 2 Report

The review reports a panoramic study about fermented foods and their role in respiratory health. The review can be of good interest for the reader of the journal but the methodological approach appears as poor clear and there are different mistake about the problem. In my opinion, the paper can be accepted for publication after some changes, according to the following indications:

• In the introduction, the work is very synthetic. The aim of the review and the novelty with respect to the existing literature should be adequately discussed. To this point, I suggest to enlarge the state-of-the-art analysis by including new parts. In the phrase present in lines 74-76, the authors reported that the fermentation foods are summarized about starter cultures, techniques etc. but in the review there aren’t a good description on these points.

• In the phrase present in lines 134-135, the authors reported that in the started cultures the strain isn’t a single strain but a symbiotic of various strains or mixed cultures. In this sense, there is a very mistake; there are more paper in this sense in which it is possible to produce fermented foods by single strain (for example: “Lactic acid bacteria co-encapsulated with lactobionic acid: Probiotic viability during in vitro digestion”, “Lactic fermentation of cereals aqueous mixture of oat and rice flours with and without glucose addition”, “Lactic fermentation of cooked navy beans by Lactobacillus paracasei CBA L74 aimed at a potential production of functional legume-based foods”, “Biotechnological production of natural sweeteners and preservatives on tomato paste” etc. etc.)

• The Figure 1 reported an original scheme? If it is a scheme taken from bibliographic resource it important to explain this.

• The Figure 2 reported an original scheme? If it is a scheme taken from bibliographic resource it important to explain this.

• It is necessary a better schematization of problem.

• The conclusions are very synthetic too. I think that they should be re-written by indicating the main conclusions achieved in this study in very clear form.

Author Response

The review reports a panoramic study about fermented foods and their role in respiratory health. The review can be of good interest for the reader of the journal but the methodological approach appears as poor clear and there are different mistake about the problem. In my opinion, the paper can be accepted for publication after some changes, according to the following indications:

Comment: In the introduction, the work is very synthetic. The aim of the review and the novelty with respect to the existing literature should be adequately discussed. To this point, I suggest to enlarge the state-of-the-art analysis by including new parts. In the phrase present in lines 74-76, the authors reported that the fermentation foods are summarized about starter cultures, techniques etc. but in the review there aren’t a good description on these points.

Author response: The whole manuscript has been reorganized and detailed for better clarification in the revised manuscript.

Comment: In the phrase present in lines 134-135, the authors reported that in the started cultures the strain isn’t a single strain but a symbiotic of various strains or mixed cultures. In this sense, there is a very mistake; there are more paper in this sense in which it is possible to produce fermented foods by single strain (for example: “Lactic acid bacteria co-encapsulated with lactobionic acid: Probiotic viability during in vitro digestion”, “Lactic fermentation of cereals aqueous mixture of oat and rice flours with and without glucose addition”, “Lactic fermentation of cooked navy beans by Lactobacillus paracasei CBA L74 aimed at a potential production of functional legume-based foods”, “Biotechnological production of natural sweeteners and preservatives on tomato paste” etc. etc.)

Author response: Thank you very much for the correction. The phrase has been removed and correct statements have been replaced in the revised manuscript.

Comment: The Figure 1 reported an original scheme? If it is a scheme taken from bibliographic resource it important to explain this.

Author response: As per Reviewer 1 suggestions, part 5 (and references cited in part 5 and figure 1) has been removed in the revised manuscript.

Comment: The Figure 2 reported an original scheme? If it is a scheme taken from bibliographic resource it important to explain this.

Author response: Figure 2 has been removed from the revised manuscript.

Comment: It is necessary a better schematization of problem.

Author response: The whole manuscript has been reorganized and detailed for better clarification in the revised manuscript.

Comment: The conclusions are very synthetic too. I think that they should be re-written by indicating the main conclusions achieved in this study in very clear form.

Author response: The conclusions have been rewritten in the revised manuscript.

Round 2

Reviewer 1 Report

The manuscript has been revised accordingly, thus it can be accepted.

Reviewer 2 Report

Line 26: CIDs isn't defined

Line 73: Words deletes are important because are a definition of URTIs